# IF-MoDGS: Initial-Free Monocular Dynamic Gaussian Splatting

## ABSTRACT

In the field of scene reconstruction with moving objects, recent studies have utilized 3D Gaussian Splatting (3DGS) for spatial representation. This method typically relies on camera poses and point clouds obtained through the Structure-from-Motion (SfM) algorithm. However, in scenes captured with monocular viewpoints and containing moving objects in each frame, the SfM algorithm struggles to obtain accurate camera poses and points clouds. As a result, it often either removes point clouds of dynamic objects or fails to find camera poses for each frame, thereby leading to sub-optimal rendering of dynamic scenes. We propose a novel approach, Initial-Free Monocular Dynamic Gaussian Splatting (IF-MoDGS) which does not require precomputed camera poses and point clouds in dynamic scenes with moving objects. Our approach estimates camera poses using the static background, separated from dynamic objects by a motion mask, and generates point clouds specifically for the dynamic objects. To handle dynamic objects, we define a canonical space and apply deformation to link it with each viewpoint and timestamp. Then, to improve quality in complex spatio-temporal scenes, we utilize a high-dimensional feature loss and an annealing frequency loss. Extensive experimental results demonstrate that our method can effectively render dynamic scenes without relying on precomputed camera poses and point clouds, achieving the state-of-the-art performance in dynamic scene rendering tasks using a monocular camera. Our project will be available at:https://anonymous.4open.science/w/IF-MODGS-67F5/

## 1 INTRODUCTION

3D scene reconstruction studies have been a significant topic in the field of computer vision. However, these tasks often face considerable limitations due to the need for extensive manpower, high computational resources, and specialized equipment. As a result, various approaches have been explored to reconstruct 3D spaces using 2D image data obtained from cameras. Neural Radiance Field (NeRF) (Mildenhall et al., 2021), significantly highlighted the importance of spatial representation. It constructs a continuous volumetric scene function through a simple multi-layer perceptron to synthesize new views of complex scenes, and it leads to numerous follow-up studies (Barron et al., 2021; 2022; 2023). Nevertheless, its slow training and rendering speeds have restricted its practical applications. Subsequently, the 3D Gaussian Splatting (3DGS) (Kerbl et al., 2023) was proposed to reconstruct 2D space by optimizing the parameters of explicit 3D Gaussians and accumulating them. This leads to a significant improvement in training and rendering speeds for 3D reconstruction tasks, making it easier to restore high-quality 3D scenes.

Since most real-world captured sequences involve dynamic scenes, reconstructing these scenes remains a significant challenge. While the reconstruction of static scenes is well-developed (Yu et al., 2024; Lin et al., 2024; Huang et al., 2024a; Hamdi et al., 2024; Lu et al., 2024a), transitioning to dynamic scene reconstruction presents numerous issues, such as the disappearance of dynamic objects or overfitting to training views. In static scenes or those captured at a single time step, models can easily understand the scene because the objects maintain consistent shapes across different viewpoints. However, in dynamic scenes, the shapes change with each time step, complicating the model's understanding of the overall scene. To address this, multi-view dynamic approaches (Wu et al., 2024; Yang et al., 2024; Lu et al., 2024b; Li et al., 2024; Sun et al., 2024; Guo et al., 2024) capture synchronized data from multiple cameras, while monocular dynamic approaches (Liang et al.,

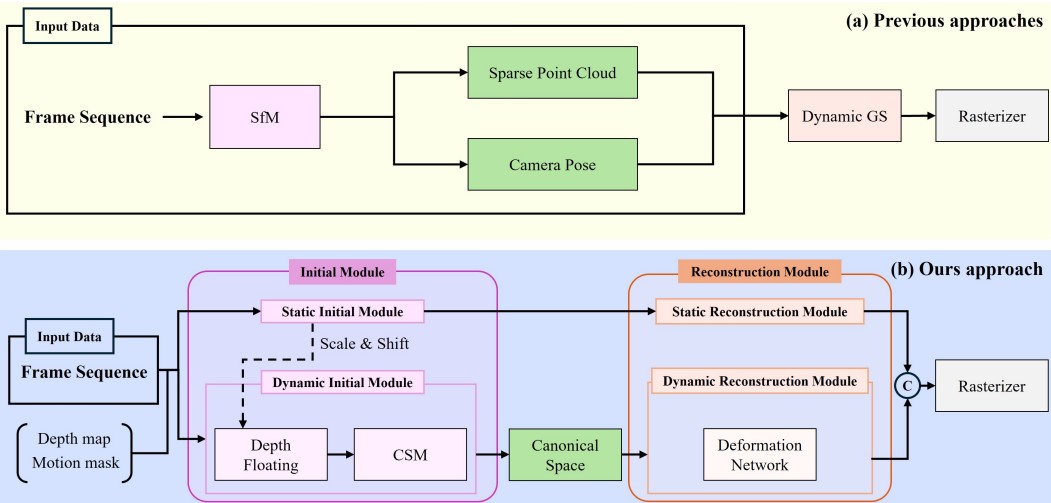

Figure 1: **Illustration of differences from existing methods.** (a) 3D Gaussian-based methodology for traditional dynamic scenes relying on SfM algorithm base initial. Since these methods are based on the SfM algorithm, there is no point for dynamic objects, and camera poses are also required as inputs. (b) Our method that works without initials.

2023; Shih et al., 2024; Huang et al., 2024b; Guo et al., 2024) reconstruct continuous scenes using a single camera. Various efforts have been made to move beyond traditional static environments, resulting in numerous studies showing high performance in reconstruction metrics. However, these methods still rely on external programs to separately obtain camera parameters and the initial point cloud. As illustrated in Figure 1(a), obtaining camera parameters and initial point clouds, which are crucial priors in 3DGS, remains difficult in dynamic scenes.

As seen in Figure 1(a), most 3D reconstruction studies have utilized external programs based on Structure from Motion (SfM) to estimate camera poses and obtain an initial point cloud before proceeding with reconstruction. However, challenges arise when acquiring camera parameters from image collections that include dynamic objects. SfM typically extracts keypoints from the images during feature extraction, and then, adjusts the feature correspondences between images using the epipolar geometry. It then estimates 3D coordinates through triangulation and minimizes errors using the bundle adjustment and outlier filtering. While dynamic features can be ignored and only static features are utilized during the outlier filtering process, which allows for the acquisition of camera parameters, this approach leads to the inability to acquire point clouds for dynamic objects due to the exclusion of unmatched dynamic points. If the dynamic region occupies a large portion of each frame, the confusion can occur during the filtering process, making it difficult to obtain both camera parameters and point cloud results. Unlike NeRF-based approaches, where the camera poses are crucial, 3DGS based methods require both the camera poses and the initial point cloud as important priors, thereby complicating the reconstruction of dynamic scenes.

To address these issues, we propose a new method called Initial-Free Monocular Dynamic Gaussian Splatting (IF-MoDGS). It enables the simultaneous estimation of both camera position and initial point cloud from dynamic scenes captured by single RGB camera, facilitating 3D reconstruction even in dynamic environments. As seen in Figure 1(b), we utilize a motion mask to separate static and dynamic regions and explicitly distinguish each region for training. The Static Initial Module (SIM) estimates the 3D points corresponding to static regions and the camera poses. For dynamic regions, the Dynamic Initial Module (DIM) estimates point clouds for dynamic objects using depth floating and the Canonical Space Mapper (CSM). The floating points are transformed into canonical coordinates through the CSM, resulting in a canonical form that has positions and shapes applicable to the entire scene. Subsequently, the Dynamic Reconstruction Module (DRM) employs a deformation network to refine the detailed shape and position according to each viewpoint and temporal information. Through this approach, we can clearly distinguish the Gaussians representing static regions from those representing dynamic objects. The separating loss enables us to optimize the Static Reconstruction Module (SRM) and DRM independently. When integrating these distinct Gaussians

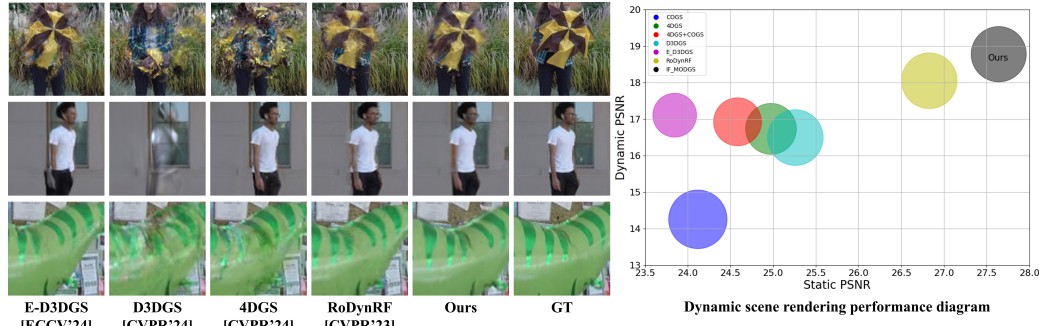

Figure 2: **Performance comparison with other dynamic scene rendering model.** The performance diagram present the mPSNR results for both static and dynamic regions, for the test view rendering outputs of each model. It demonstrates that our model outperforms existing models in both static and dynamic areas.

to form a unified scene, we use a VGG loss at the high-dimensional feature level to ensure proper alignment and an annealing frequency loss to effectively manage the complex spatio-temporal reconstruction of dynamic objects.

Our contributions are as follows:

- We propose a novel approach, IF-MoDGS, enabling the high-quality novel view rendering for dynamic scenes captured by single RGB camera without requiring initial camera poses and point clouds.
- We estimate 3D points for moving objects by reconstructing dynamic objects through the canonical space, overcoming challenges faced by previous methods. To handle the canonical space, we employ the CSM along with deformation to transform and align dynamic objects across different time steps. Through this approach, we explicitly separate dynamically changing objects in complex spatio-temporal scenes from the static regions, enabling effective reconstruction of the dynamic areas.
- We optimize Gaussians for the static and dynamic regions separately by applying a mask-based separating loss. This independent optimization of the SRM for static scenes and the DRM for dynamic objects enhances the rendering quality of both components.

As a result, our method experimentally demonstrated the ability to reconstruct dynamic scenes using only single RGB camera and achieved high performance in novel view synthesis, as illustrated in Figure 2, for both dynamic and static regions.

## 2 RELATED WORKS

### 2.1 INITIAL-FREE 3D RECONSTRUCTION

NeRF (Mildenhall et al., 2021) generates a continuous volumetric scene function using an MLP, focusing on novel view synthesis rather than explicit 3D reconstruction. However, its reliance on MLP makes it prone to overfitting, sensitive to noise, and difficult to edit. Unlike NeRF-based methods (Mildenhall et al., 2021; Barron et al., 2021; 2022; 2023; Liu et al., 2020; Yu et al., 2021b; Chen et al., 2022; Yu et al., 2021a), which need to compute large volumes of empty space, 3DGS employed a more efficient computation, offering faster rendering times by reducing unnecessary computations in empty space. Despite these advances, both methods still rely heavily on accurate camera parameters and initial point clouds. Recent efforts to overcome these limitations include NeRF-based methods like BARF (Lin et al., 2021a) and RoDyNRF (Liu et al., 2023), which estimate camera poses and reconstruct scenes using only RGB images. BARF focuses on static scenes, jointly optimizing both camera poses and an MLP by calculating optical flow and using the Lucas-Kanade algorithm to estimate the displacement vectors of pixel coordinates. RoDyNRF, on the other hand, distinguishes between static and dynamic pixels, adjusting alignment of back-projected points.

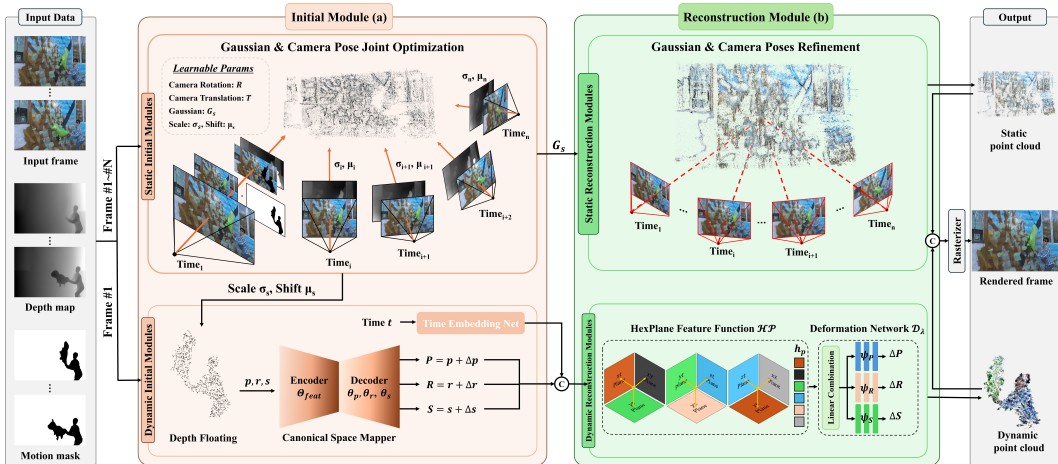

Figure 3: **The overall pipeline of our method.** It consists of two main parts: the initial module and the reconstruction module. The initial module jointly optimizes the Gaussian and camera poses, and the reconstruction module refines the camera poses and point cloud. We then render static and dynamic separately to allow for optimizations specific to each.

For dynamic regions, RoDyNRF uses a scene flow MLP to estimate the motion of 3D points. As a result, it successfully reconstructs dynamic scenes and estimates camera poses using only image data from a single camera. In 3DGS-based research, accurate camera poses and initial point clouds remain crucial. Approaches like Colmap-free 3DGS (Fu et al., 2024) and COGS (Jiang et al., 2024) aim to estimate poses using monocular depth and sparse image sets, but prior information is still a key constraint. Our proposed IF-MoDGS addresses these issues by estimating camera poses and reconstructing both static and dynamic scenes. We separate Gaussians representing static and dynamic regions, optimizing camera poses from static regions and reconstructing dynamic regions through a learned canonical form. This method allows for reconstruction using only image data, overcoming the limitations of prior 3DGS methods.

## 2.2 DYNAMIC RENDERING

Unlike static scene reconstruction, dynamic scene reconstruction is a very challenging task as it involves compositing the desired viewpoint at a specific time. Dynamic scenes can be acquired in two ways: from multi-view cameras (Fridovich-Keil et al., 2023; Cao & Johnson, 2023; Wang et al., 2023) or from monocular cameras (Liang et al., 2023; Chen et al., 2024). In addition to this, there are other assumptions, such as whether the camera positions change over time, or whether RGB-D cameras are used, but in general, the dataset is constructed in the same way. In the case of multi-view cameras, there are multiple images for each time step, so there is enough information to use SfM and other methods. However, if we consider the case of a monocular camera, there is only one scene captured at each time step. In the case of dynamic objects, the shape and position change from time to time, so it is very difficult for the model to understand the reconstruction using only that information. Most of the readily accessible videos are single-camera and were recorded in dynamic scenes. Therefore, it can be seen that dynamic scene reconstruction is a very important task. In 3DGS, study of monocular dynamic scene reconstruction exists (Yang et al., 2024). This study utilizes a scenario where a single camera is used, but it has the disadvantage of using point clouds acquired using stereo depth. In the case of (Kocabas et al., 2024), the dynamic objects are limited to humans and an additional initial prior called SMPL (Loper et al., 2023) is provided. We overcome the aforementioned challenges in monocular dynamic scenes. In a monocular view with moving objects, we used video frames and corresponding monocular depth to estimate the correct camera pose. Through backprojection, We generate an initial prior for unspecified dynamic objects, rather than focusing on a specific dynamic object as SMLP does.

## 3 METHOD

Our IF-MoDGS reconstructs and renders dynamic scenes without initial priors. IF-MoDGS consists of the initial module that estimates the initial 3D points and approximates camera poses, and the reconstruction module that optimizes Gaussians using the initial points and estimates the exact camera poses as shown in Figure 3. The initial module consists of the SIM, which estimates the initial 3D points and camera poses for static scene, and the DIM, which estimates the 3D points for dynamic objects. The reconstruction module consists of the SRM and the DRM. The SRM takes the initial 3D points and camera poses estimated by the SIM as inputs, optimizes the Gaussians, and refines the camera poses for greater accuracy. The DRM takes as input the initial 3D points estimated by the DIM and optimizes the Gaussians and deformation. The final rendering result can be obtained by concatenating the Gaussian from the SRM and DRM.

### 3.1 PRELIMINARY

**3D Gaussian Splatting** 3D Gaussian Splatting (Kerbl et al., 2023) represents a 3D scene by defining a 3D Gaussian for each point in a point cloud. Each 3D Gaussian is a distribution defined by a mean vector $\mu \in \mathbb{R}^3$ and an anisotropic covariance matrix $\Sigma \in \mathbb{R}^{3\times3}$, forming an ellipsoid. The 3D Gaussian is mathematically defined as follows:

$$G(\mathbf{x}) = e^{-\frac{1}{2}(\mathbf{x}-\mu)^T \Sigma^{-1}(\mathbf{x}-\mu)}, \tag{1}$$

where the mean vector $\mu$ corresponds to the 3D coordinate of the point cloud. The covariance matrix $\Sigma$ of the 3D Gaussian if further decomposed into a rotation matrix $R \in \mathbb{R}^{3\times3}$ and a scaling vector $S \in \mathbb{R}^3$ of the ellipsoid, expressed as:

$$\Sigma = RSS^T R^T. \tag{2}$$

The 3D Gaussian is projected into 2D space corresponding to each camera view with the covariance of the rendered 2D Gaussian described as follows:

$$\Sigma' = JW\Sigma W^\top J^\top, \tag{3}$$

where $J$ is the Jacobian matrix of the affine approximation for the projective transformation, and $W$ denotes the transformation matrix from world coordinates to camera coordinates. For final image rendering, the 3D Gaussian is parameterized by its position $\mathbf{x} \in \mathbb{R}^3$, opacity $o \in \mathbb{R}$, and color $c \in \mathbb{R}^k$, where color is expressed through spherical harmonics coefficients. These 2D Gaussians are rasterized similarly to the volume rendering of NeRF (Mildenhall et al., 2021) technique to determine the final pixel values. The rasterization process that computes the final pixel values is performed using alpha-blending, as follows:

$$C(u,v) = \sum_{i \in N} T_i \alpha_i c_i, \quad \alpha_i = o_i e^{-\frac{1}{2}(\mathbf{x}-\mu_i)^\top \Sigma'(\mathbf{x}-\mu_i)}, \quad T_i = \prod_{j=1}^{i-1}(1-\alpha_j) \tag{4}$$

where $T_i$ is the transmittance and $\mu_i$ represents the $uv$ coordinate of the 3D Gaussians projected onto the 2D image plane.

### 3.2 INITIAL MODULE

We propose an initial module that estimates the initial 3D points needed for Gaussian optimization, and in Figure 3(a), the scene is separated into the static scene and the dynamic objects using a motion mask. The approximate camera poses and 3D points can be obtained from the initial module.

**Static Initial Module** The SIM, utilizing the 3D Gaussian learning methodology of COGS(Jiang et al., 2024), reconstructs the initial 3D points and obtains an approximate camera poses, focusing on the static scene. To focus solely on the static scene, we use a motion mask $M$ generated by the Mask R-CNN (He et al., 2017) to exclude moving objects from the scene. From the video frame $I(u,v)_i$ and the corresponding monocular depth (Ke et al., 2024) $D_i$, the initial 3D points are obtained as follows by back-projecting each pixel onto 3D space, which can be expressed as:

$$\mathbf{p}_i = [R|t]_i \cdot \left( \hat{d}_i(u,v) \cdot M_i(u,v) \cdot K_i^{-1} \begin{pmatrix} u \\ v \\ 1 \end{pmatrix}_i \right), \tag{5}$$

where $p_i$ is the back-projected 3D point of the $i$-th view. The optimization to align the consistency between views by adjusting the camera pose and depth information is performed as follows:

$$\min_{R,t} \| M_i \odot I_i - M_i \odot \hat{I}_i \|_1, \tag{6}$$

where $\odot$ denotes elementwise multiplication, $\hat{I}_i$ is the rendered image and $[R|t]_i$ is the camera pose of view $i$. In Figure 3(a), the depth value for each view is scaled by the scale parameter $\sigma_i$ and the shift parameter $\mu_i$, which are learned jointly with the camera poses as described below:

$$\hat{d}_i(u,v) = D_i(u,v) \times \sigma_i + \mu_i, \tag{7}$$

This allows us to remove moving objects that interfere with camera pose estimation and 3D Gaussian learning, and subsequently reconstruct the camera poses and 3D points.

**Dynamic Initial Module** As the motion mask is applied in the SIM, information about the dynamic object is lost. We use an inverse motion mask$(1 - M)$ to reconstruct dynamic object information. A 3D point is initialized using the first video frame and the depth, which is scaled and shifted in the SIM. The process for floating the 3D point is shown below:

$$\mathbf{p}_w = [R|t]_0 \cdot \left( \hat{d}_0(u,v) \cdot (1 - M_0(u,v)) \cdot K_0^{-1} \begin{pmatrix} u \\ v \\ 1 \end{pmatrix} \right), \tag{8}$$

where $K_0$ is the intrinsic camera parameter matrix for the first frame, and $u,v$ represent the pixel coordinates of the frame. $[R|t]_0$ denotes the camera pose, and $\hat{d}_0(u,v)$ is the depth value, adjusted by the depth scale and shift estimated from the SIM to ensure consistency of the monocular depth for the first view. The dynamic point cloud obtained from the above process is unstable due to adjustments made with the static scale and shift values. To address this, we applly quantile sampling to remove the top 5% and bottom 10% of points, followed by the point-wise interpolation to convert sparse points into dense points. The resulting densified points are as follows:

$$\mathbf{p}_{\text{dynamic}} = \mathbf{p}_w \oplus \text{interp} \left[ \text{quantile}(\mathbf{p}_w) \right], \tag{9}$$

where $\oplus$ represents the operation of combining the original points with the interpolated values from the quantile sampling. The resulting densified points $\mathbf{p}_{\text{dynamic}}$ are represented as 3D Gaussians $\mathbf{G} = \{p, r, s\}$, where $p$ is the 3D point coordinate vector derived from $\mathbf{p}_{\text{dynamic}}$, $r$ is the rotation matrix, and $s$ is the scale vector.

3D points floated using depth information still exist in incomplete locations due to inaccuracies caused by the scaling and shifting from the static depth. Specifically, dynamic objects tend to be incorrectly positioned too close to the background. To resolve this, the 3D points are initially floated using depth information and, after refinement through the quantile sampling and interpolation, they are transformed into the scaled canonical space coordinate via the CSM network. The CSM network is crucial for adjusting the inaccurate positions of 3D points, as it corrects the errors resulting from the static depth scaling and shifting. The encoder $\mathbf{E} = \Theta_{feat}$ of the CSM network is responsible for extracting features from the 3D points. The decoder $\mathbf{D} = \{\theta_p, \theta_r, \theta_s\}$ of the CSM network consists of three networks that estimate the position offset vector, rotation offset vector, and scale offset vector, as follows:

$$\Delta \mathbf{p} = \theta_P(\Theta_{feat}(p)), \quad \Delta \mathbf{r} = \theta_r(\Theta_{feat}(p)), \quad \Delta \mathbf{s} = \theta_s(\Theta_{feat}(p)). \tag{10}$$

The offset vectors output by the CSM network move the 3D Gaussians $\mathbf{G}$ to the canonical space, forming the 3D Gaussians $\mathbf{G_{can}}$ in the canonical space as follows:

$$\mathbf{G_{can}}\{p', r', s'\} = \Delta \mathbf{G} + \mathbf{G} = \{p + \Delta \mathbf{p}, r + \Delta \mathbf{r}, s + \Delta \mathbf{s}\}, \tag{11}$$

where $\mathbf{G_{can}}\{p', r', s'\}$ is the final canonical Gaussian constructed through the CSM network.

### 3.3 RECONSTRUCTION MODULE

We propose a reconstruction module where the Gaussians, which are separated into static and dynamic, are optimized individually. For static Gaussians, parameters are learned to reconstruct them in greater detail, and the camera poses from the initial module is further adjusted for greater accuracy. In the dynamic regions, the initial Gaussians for dynamic objects, transferred to canonical space, are modified at each time step using a Deformation Network $\mathcal{D}_\lambda$.

**Static Reconstruction Module** For the results of the initial module, we obtain explicitly separated Gaussians for static and canonical Gaussians for dynamic object. In the SRM, only the optimization of the Gaussian corresponding to the static and the adjustment of the camera poses are done. When optimizing the Gaussian, we freeze the gradient for the camera poses, and when optimizing the camera pose, we freeze all parameters other than the camera poses and apply only the photometric loss, as detailed in the Training setup section of the appendix. The Gaussian corresponding to static is optimized by separating losses, which is described in detail in the section of Rendering and Objective Function.

**Dynamic Reconstruction Module** The DRM converts the shape of the Gaussians $G_{can} = \{p', r', s'\}$ in canonical space into real space for each time. We extract the hex-plane feature $\mathbf{h_p}$ for the position $p'$ of the canonical space Gaussian $G_{can} = \{p', r', s'\}$ and the time embedding $t$, following the method used in 4DGS (Wu et al., 2024):

$$\mathbf{h_p} = \mathcal{HP}(PE(p'), PE(t)), \tag{12}$$

where $PE$ is the positional encoding, and $\mathcal{HP}$ represent hex-plane feature function for position and time. The deformation network $\mathcal{D}_\lambda$ uses $h_p$ to output the parameters $(\Delta p', \Delta R, \Delta S)$ of the deformed Gaussian:

$$\Delta G_{can}\{\Delta P, \Delta R, \Delta S\} = \mathcal{D}_\lambda(h_p). \tag{13}$$

The hex-plane features are combined and passed through the decoder to estimate the offset vectors:

$$\Delta P = \Psi_P(LC(h_p)), \quad \Delta R = \Psi_R(LC(h_p)), \quad \Delta S = \Psi_S(LC(h_p)), \tag{14}$$

where $LC$ is the linear combination layer. The final 3D Gaussian in the real space is represented as:

$$G_{real}\{P_{real}, R_{real}, S_{real}\} = \Delta G_{can} + G_{can}, \tag{15}$$

where $G_{real}$ is the 3D Gaussian for a dynamic object deformed in the real space.

## 3.4 RENDERING AND OBJECTIVE FUNCTION

**Rendering** From the reconstruction module results, we obtain 3D Gaussians for both static and dynamic components, explicitly separated in the real space. This separation allows for independent rendering of both the static and dynamic components. The rendering equation is the same as the rendering through alpha-blending in 3D Gaussian Splatting (Kerbl et al., 2023).

$$I_{Static}(u,v) = \sum_{i=1}^{N} p_{i,S}(\mu_S, \Sigma_S)\alpha_{i,S}c_{i,S} \prod_{j=1}^{i-1}(1 - p_{i,S}(\mu_S, \Sigma_S)\alpha_{i,S}c_{i,S}), \tag{16}$$

where $p_{i,S}$ is the i-th Gaussian corresponding to the static part in real space, and $\mu_S$ and $\Sigma_S$ are the mean and covariance of the Gaussian, respectively. $\alpha$ is computed based on the opacity of the Gaussian, and $c_S$ is the color value calculated using spherical harmonics.

$$I_{Dynamic}(u,v,t) = \sum_{i=1}^{N} p_{i,D}(\mu_{t,D}, \Sigma_{t,D})\alpha_{i,D}c_{i,D} \prod_{j=1}^{i-1}(1 - p_{i,D}(\mu_{t,D}, \Sigma_{t,D})\alpha_{i,D}c_{i,D}), \tag{17}$$

where it has the same form as the static, but with parameters corresponding to the dynamic Gaussian at time $t$. By concatenating the Gaussians corresponding to the static and dynamic components, we can obtain the Gaussians for the entire scene. By rendering the Gaussians for the entire scene, the final rendering result can be obtained as shown below:

$$I_{Fusion}(u,v,t) = \sum_{i=1}^{N} p_i(\mu_t, \Sigma_t)\alpha_i c_i \prod_{j=1}^{i-1}(1 - p_i(\mu_t, \Sigma_t)\alpha_i c_i), \tag{18}$$

where all parameters for the Gaussians are the concatenated parameters of the static and dynamic components.

**Objective function** We propose a separating loss function that utilizes images rendered with static and dynamic separation, as shown in the appendix. If the photometric loss is applied solely to the final image, rendered from the concatenation of static and dynamic components, each Gaussian will not be optimized independently. The static Gaussians densify and fill the dynamic regions,

Table 1: Comparisons of the results for the overall average and quantitative table for 7 of the NVIDIA dataset. We report PSNR, LPIPS and color each cell as **best** and **second best**

| Methods | Jumping PSNR↑ | LPIPS↓ | Skating PSNR↑ | LPIPS↓ | Truck PSNR↑ | LPIPS↓ | Umbrella PSNR↑ | LPIPS↓ | Balloon1 PSNR↑ | LPIPS↓ | Balloon2 PSNR↑ | LPIPS↓ | Playground PSNR↑ | LPIPS↓ | Avg PSNR↑ | LPIPS↓ |
|---|---|---|---|---|---|---|---|---|---|---|---|---|---|---|---|---|
| COGS | 20.72 | 0.211 | 23.53 | 0.127 | 22.72 | 0.202 | 20.06 | 0.251 | 19.22 | 0.233 | 23.75 | 0.140 | 20.67 | 0.111 | 21.61 | 0.182 |
| 4DGS | 21.98 | 0.178 | 26.53 | 0.072 | 24.45 | 0.128 | 21.77 | 0.183 | **21.22** | **0.154** | 24.73 | 0.079 | 20.97 | 0.095 | 23.00 | 0.127 |
| 4DGS+COGS | 22.81 | 0.138 | 25.03 | 0.104 | 25.67 | **0.055** | 22.02 | 0.152 | 20.05 | 0.175 | **25.41** | **0.067** | 19.29 | 0.116 | 22.90 | 0.115 |
| D3DGS | 21.33 | 0.213 | 24.77 | 0.132 | 25.33 | 0.120 | 22.30 | 0.133 | 21.02 | 0.144 | 24.63 | 0.078 | 22.11 | 0.075 | 23.01 | 0.128 |
| E-D3DGS | 21.47 | 0.158 | 25.70 | 0.078 | 25.97 | 0.087 | 21.62 | 0.166 | 19.88 | 0.166 | 23.53 | 0.091 | 19.34 | 0.133 | 22.50 | **0.098** |
| RoDynRF (**w/o COLMAP**) | **23.73** | **0.100** | **29.22** | **0.045** | **29.00** | 0.075 | **23.38** | **0.115** | 21.57 | 0.137 | 20.94 | 0.169 | **24.37** | **0.067** | **24.60** | 0.101 |
| IF-MODGS (**Ours w/o initial**) | **23.45** | **0.104** | **28.30** | **0.046** | **28.10** | **0.071** | **23.33** | **0.099** | **23.24** | **0.092** | **26.42** | **0.045** | **24.92** | **0.034** | **25.37** | **0.070** |

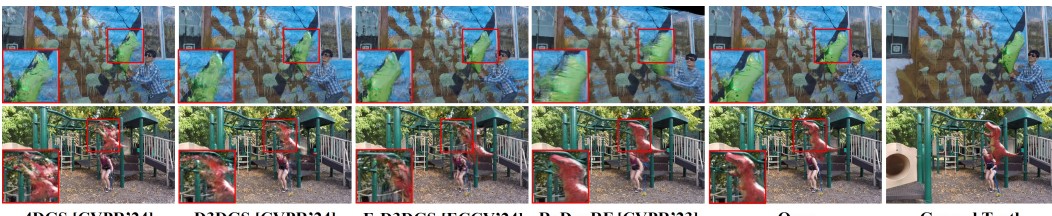

**4DGS [CVPR'24]**  **D3DGS [CVPR'24]**  **E-D3DGS [ECCV'24]**  **RoDynRF [CVPR'23]**  **Ours**  **Ground Truth**

Figure 4: **Qualitative results for Balloon2, Playground scene of NVIDIA dataset.** It shows the clearest novel view rendering result for a moving object compared to other methods.

while the dynamic Gaussians are rendered in the static areas, ultimately resulting in the issue where no movement occurs. The separating loss is calculated by rendering only the dynamic Gaussians, multiplying by the inverse motion mask, and computing the photometric loss, then rendering only the static Gaussians, multiplying by the motion mask, and combining the photometric losses. As follows:

$$Loss_{Separate} = M \cdot L1(I_{GT}, I_{Static}) + (1 - M) \cdot L1(I_{GT}, I_{Dynamic}). \quad (19)$$

By applying the separating loss, the static Gaussians are optimized for the static regions, and the dynamic Gaussians are optimized for the dynamic objects. Additionally, we calculate the photometric loss for the concatenated final image, as shown below:

$$Loss_{photo} = L1(I_{GT}, I_{Fusion}). \quad (20)$$

Both static and dynamic Gaussians are used to render the final image. To properly blend the static and dynamic Gaussians, additional training control is applied using the VGG loss and annealing frequency loss. The VGG loss measures the difference between the feature vectors of the ground truth image and the rendered image. Minimizing the loss between images solely at the pixel level such as using L1 loss, tends to produce blurred results in complex spatio-temporal situations. This is because it is more efficient for model to optimize by enlarging the region occupied by a single Gaussian and reducing significant temporal changes. Therefore, we additionally employ VGG loss to preserve the details of the final output. Annealing frequency loss was also introduced for similar reasons. It is employed to provide additional control over reconstructing high-frequency information in the scene. Below is the configuration of our final loss. The results of ablation for our methods are shown in Table 3.

$$Loss_{final} = \lambda_1 Loss_{photo} + \lambda_2 Loss_{Separate} + \lambda_3 Loss_{vgg} + \lambda_4 Loss_{Freq}, \quad (21)$$

where $\lambda_1 = 0.2, \lambda_2 = 0.8, \lambda_3 = 0.75, \lambda_4 = 0.1$.

# 4 EXPERIMENTS

Based on the used datasets and evaluation metrics, we compare our model with existing SOTA models. We used 3D Gaussian models targeting dynamic scene, models that jointly estimate camera poses and Gaussians, and models that jointly estimate NeRF-based dynamic scene and camera poses for the comparison.

Table 2: Comparison of quantitiave and average values for 6 UCSD. We report PSNR, LPIPS and color each cell as **best** and **second best**

| Methods | Jumping | | Skating | | Truck | | Umbrella | | Balloon1 | | Balloon2 | | Playground | | Avg | |
|---|---|---|---|---|---|---|---|---|---|---|---|---|---|---|---|---|
| | PSNR↑ | LPIPS↓ | PSNR↑ | LPIPS↓ | PSNR↑ | LPIPS↓ | PSNR↑ | LPIPS↓ | PSNR↑ | LPIPS↓ | PSNR↑ | LPIPS↓ | PSNR↑ | LPIPS↓ | PSNR↑ | LPIPS↓ |
| COGS | 20.72 | 0.211 | 23.53 | 0.127 | 22.72 | 0.202 | 20.06 | 0.251 | 19.22 | 0.233 | 23.75 | 0.140 | 20.67 | 0.111 | 21.61 | 0.182 |
| 4DGS | 21.98 | 0.178 | 26.53 | 0.072 | 24.45 | 0.128 | 21.77 | 0.183 | 21.22 | 0.154 | 24.73 | 0.079 | 20.97 | 0.095 | 23.00 | 0.127 |
| 4DGS+COGS | 22.81 | 0.138 | 25.03 | 0.104 | 25.67 | 0.055 | 22.02 | 0.152 | 20.05 | 0.175 | 25.41 | 0.067 | 19.29 | 0.116 | 22.90 | 0.115 |
| D3DGS | 21.33 | 0.213 | 24.77 | 0.132 | 25.33 | 0.120 | 22.30 | 0.133 | 21.02 | 0.144 | 24.63 | 0.078 | 22.11 | 0.075 | 23.01 | 0.128 |
| E-D3DGS | 21.47 | 0.158 | 25.70 | 0.078 | 25.97 | 0.087 | 21.62 | 0.166 | 19.88 | 0.166 | 23.53 | 0.091 | 19.34 | 0.133 | 22.50 | 0.098 |
| RoDynRF (w/o COLMAP) | 23.73 | 0.100 | 29.22 | 0.045 | 29.00 | 0.075 | 23.38 | 0.115 | 21.57 | 0.137 | 20.94 | 0.169 | 24.37 | 0.067 | 24.60 | 0.101 |
| IF-MODGS (Ours w/o initial) | 23.45 | 0.104 | 28.30 | 0.046 | 28.10 | 0.071 | 23.33 | 0.099 | 23.24 | 0.092 | 26.42 | 0.045 | 24.92 | 0.034 | 25.37 | 0.070 |

**4DGS [CVPR'24]**  **D3DGS [CVPR'24]**  **E-D3DGS [ECCV'24]**  **RoDynRF [CVPR'23]**  **Ours**  **Ground Truth**

Figure 5: **Qualitative results of previous methods and our method.** Previous methods fail to synthesize in novel view, but our method shows perfect novel view synthesis.

## 4.1 EXPERIMENT SETUP

### 4.1.1 DATASETS

**The Nvidia dataset** (Yoon et al., 2020) is a collection of data created by edited video captured in a multi-view camera setup, but processed under a monocular setting. The frame of each view camera is continuously connected to reproduce the moving monocular camera environment and use it as training data. The test the video from the first view camera is used as the GT. The trained model fixes the input first frame with a fix view and then renders a continuous frames for evaluation.

**The UCSD dataset** (Lin et al., 2021b), like Nvidia dataset, is a video dataset shot with a multi-view camera, edited with monocular settings and used for learning and evaluation.

## 4.2 RESULTS OF NOVEL VIEW SYNTHESIS

We compared our model with 4DGS (Wu et al., 2024), D3DGS (Yang et al., 2024), E-D3DGS (Bae et al., 2024), which are 3D Gaussian-based models targeting dynamic scenes. We also compared dynamic scenes with RoDynRF (Liu et al., 2023), a NeRF-based method that can be rendered without camera poses. Since the Gaussian-based model requires a Gaussian initial, it used the information obtained by COLMAP, and our model was driven by receiving only intrinsic as a Gaussian initial. COGS (Jiang et al., 2024), which operates without an initial for static scenes, was also compared with the model operated an initial in dynamic scenes by combining 4DGS (Wu et al., 2024) and COGS (Jiang et al., 2024). Table 1 compares the quantitative results for the NVIDIA dataset, and Table 2 compares the quantitative results for the UCSD dataset. Our method had the highest average value for quantitative results in both datasets. Figure 4 shows that models (Wu et al., 2024; Yang et al., 2024; Bae et al., 2024) using COLMAP did not have a point for dynamic objects, so the rendering results for moving objects were blurred or the shape was not completed. In Figure 5, our method maintains the shape of the moving object, but in the case of comparative models, the shape is not complete and some of them disappear. These results show that our method optimizes and renders Gaussians well for the dynamic scene without initialization.

## 4.3 ABLATION STUDIES

**Dynamic initial point** Existing methods (Wu et al., 2024; Yang et al., 2024; Bae et al., 2024) fill the dynamic area through densification using static point cloud or create a point by using a

Table 3: Ablation studies on Nvidia dataset using our proposed method. The **best** results are denoted by bold

| Methods | Truck | | Balloon2 | | Avg (7 scene) | |
|---|---|---|---|---|---|---|
| | PSNR↑ | LPIPS↓ | PSNR↑ | LPIPS↓ | PSNR↑ | LPIPS↓ |
| w/o Dynamic initial points | 26.10 | 0.060 | 25.20 | 0.073 | 23.68 | 0.111 |
| w/o CSM | 27.50 | 0.076 | **26.64** | 0.046 | 25.20 | 0.073 |
| w/o Separating Loss | 23.64 | 0.142 | 24.72 | 0.081 | 23.39 | 0.114 |
| w/o Feature Loss | 27.24 | 0.107 | 26.05 | 0.088 | 25.27 | 0.105 |
| IF-MODGS (**Ours**) | **28.10** | **0.071** | 26.42 | **0.045** | **25.37** | **0.070** |

random point. However, we use a floating method that reconstructs only points corresponding to dynamic objects using a monocular view. In Section 3.2, we discussed how to initialize the points corresponding to dynamic objects using monocular depth. In Table 3, PSNR represents rendering performance decreased by 6.6% when an initial point is randomly generated and used without the relevant content. Randomly generated points contain very few useful points, as many are located far from the expected position of the real object. In addition, it took a long time to converge and was easy to fall into the local minimum because the points have to learn the movement together while matching the shape.

**Canonical Space Mapper** We used the Canonical Space Mapper(CSM) to construct canonical spaces well. Even if the point for the dynamic object is reconstructed through depth floating, there is a limitation in considering that an accurate canonical space is configured. This is because the scale shift of the depth corrected in the static area is used. Thus, CSM is used to correct the inaccurate depth alignment. The point of the dynamic object corrected up to the depth alignment constitutes a more accurate canonical space. Table 3 shows that CSM better structured the canonical space by decreasing the PSNR performance by 0.67% when CSM was not applied.

**Separating Loss** We applied a separating loss to independently optimize 3D Gaussian corresponding to static scene and 3D Gaussian corresponding to dynamic object. If the separating loss is not applied, the 3D Gaussians corresponding to the static scene will become densified and fill the regions of dynamic objects, or the 3D Gaussians corresponding to dynamic objects will fill the static scene. In such cases, the dynamic object ends up remaining still without movement, or the final output will produce a noisy rendering of the static scene. In Table 3, it can be seen that the PSNR decreases by 7.8% when the Separating Loss is not applied.

**Feature Loss** We made the rendering results clear by using a feature loss using a pre-trained VGG network. A photometric loss, which supervises only the difference between pixels, has limitations in completing the temporal change and shape of 3D Gaussian. The contents of feature loss were discussed in section 3.4. By using the feature of the rendered image, the texture of the dynamic object may be clarified, and the 3D Gaussian may be densely cloned in the learning process. Finally, in Table 3, the effect on feature loss is confirmed by decreasing the PSNR value by 0.39%.

## 5 CONCLUSION

We proposed a new method, IF-MoDGS, in Gaussian Splatting for dynamic scenes. Unlike other methodologies that relied on SfM-based algorithms like COLMAP to obtain initial values, we operated without initial point clouds and camera poses. Our method separated static and dynamic components to generate the initial Gaussian points. The initial point was reconstructed using the monocular depth, and the depth of the dynamic region with incomplete consistency was compensated using the CSM. It also improved rendering quality by optimizing the Gaussian for static and dynamic components independently. Finally, we applied a feature loss to induce the segmented static and dynamic Gaussians to be well combined and rendered. We demonstrated the validity of our new method by achieving SOTA performance on a dataset targeting monocular dynamic scenes.

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

## A  IMPLEMENTATION DETAILS

### A.1  TRAINING SETUP

We took the camera intrinsic parameter as input and used it. Also, using the pre-trained relative depth estimation model (Ke et al., 2024) and Mask R-CNN (He et al., 2017), we use depth maps and motion masks. Our method performs back-propagation of gradients for camera poses only on the SRM to jointly learn camera poses and 3D Gaussian, and does not learn gradients for camera poses on the DRM. In addition, when learning 3D Gaussian and deformation, the next iteration performs iteration that learns only camera poses in the SRM, as shown below:

$$L_{cam} = L1_{static} = (I_{static} - I_{GT}) \cdot M, \tag{22}$$

where $I_{static}$ is the render result of the SRM, and $I_{GT}$ is the Ground Truth image. The motion mask $M$ is applied pixel-wise to suppress the effect of the Gaussian on the dynamic object. And the next iteration repeats the process of learning 3D Gaussian and deformation again. Gaussian rendering used COGS (Jiang et al., 2024) rasterizer, in which the gradient calculation for the camera poses are implemented as CUDA. The Canonical Mapping Network consists of MLP. Both encoder and decoder use the softplus function as an activation function, and each layer has a width of 256. In the case of the scale decoder, it is output through the sigmoid function, and the output of the rotation decoder is normalized.

### A.2  CANONICAL SPACE MAPPER

The CSM consists of two main components: an CSM encoder $\mathbf{E} = \Theta_{feat}$ and a CSM decoder $\mathbf{D} = \{\theta_p, \theta_r, \theta_s\}$. In the CSM encoder, the information of location is expanded using positional embedding as used in NeRF (Mildenhall et al., 2021) and passed through two linear layers. The extracted features are then passed through the CSM decoder $\mathbf{D}$, which has four linear layers. At this time, first layer of the CSM decoder $\mathbf{D}$ fuses the feature dimensions using skip-connection to prevent the loss of location information. Finally, the generalized canonical field is constructed by predicting the deformation in position, scale, and rotation. The overall network structure is shown in Figure 6

### A.3  EVALUATION METRICS

PSNR and LPIPS are used as metrics to evaluate the performance of the model. PSNR is an metric that measures how the pixel values are the same between images. LPIPS is a metric that measures the similarity between features of two images using layers of a pre-trained network. We leverage the pre-trained AlexNet for the metric of LPIPS.

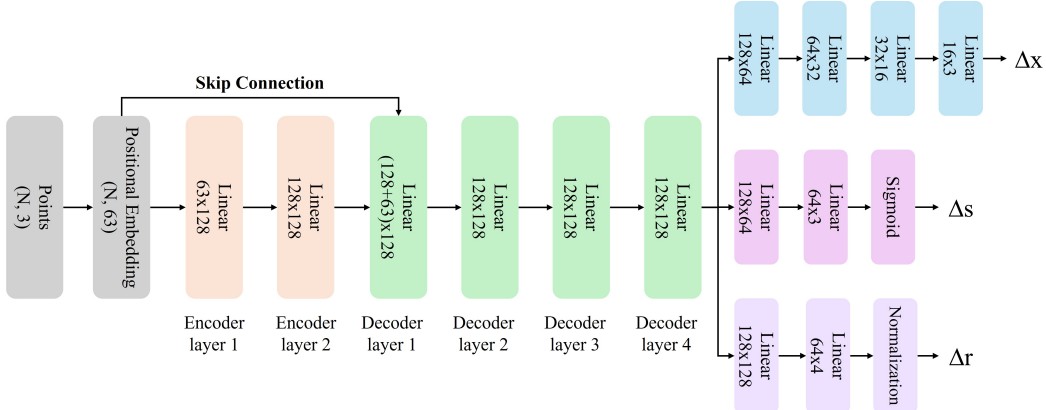

Figure 6: **Overall archtecture of the Canonical Space Mapper which map to canonical field using MLP layers.**

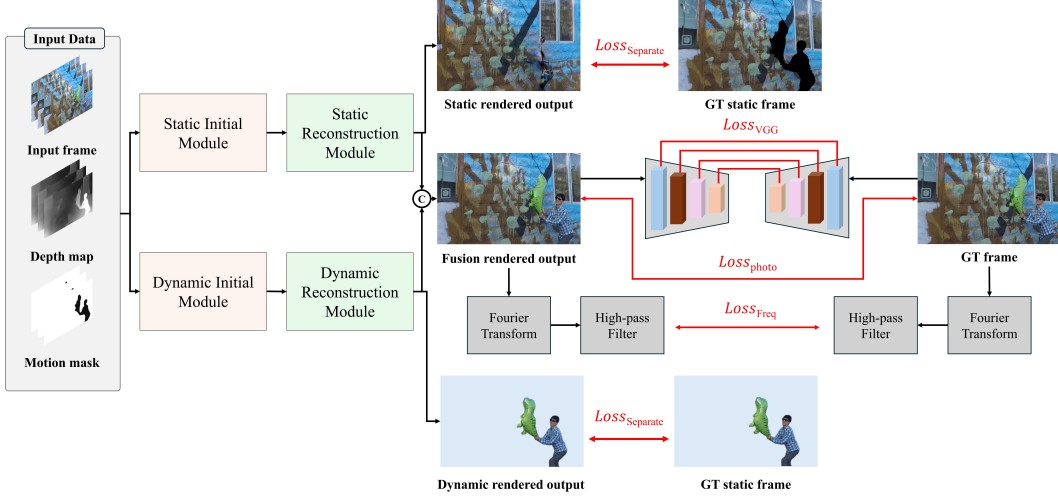

Figure 7: **Overall pipeline of our method's objective function during training.**

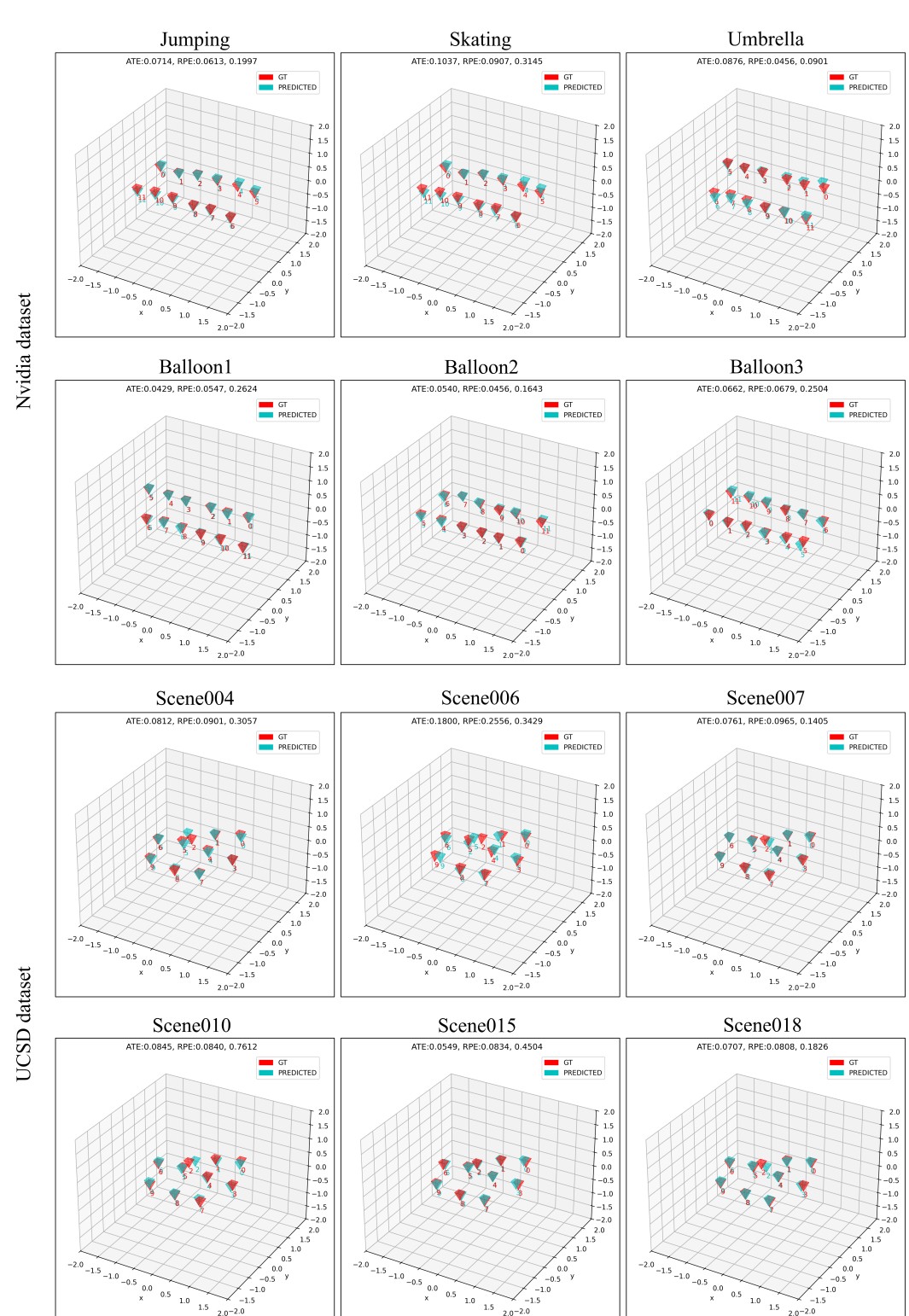

Figure 8: **Qualitative results of moving camera location on Nvidia dataset.**

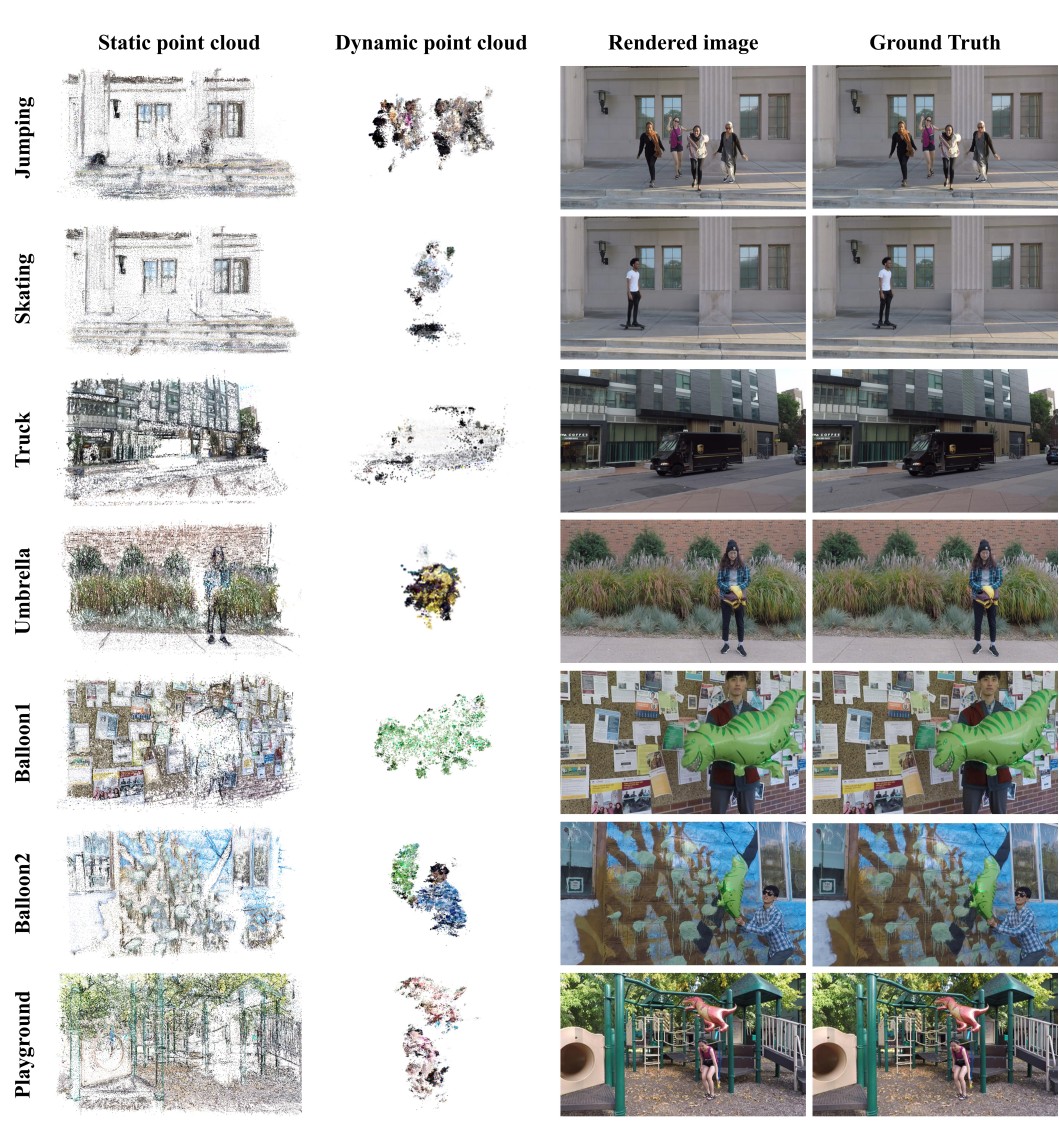

Figure 9: **Qualitative results of point cloud and rendered images on Nvidia dataset.**

