# OpenReview forum: "IF-MODGS : INITIAL FREE MONOCULAR DYNAMIC GAUSSIAN SPLATTING"
_ICLR.cc/2025/Conference — ICLR 2025 Conference Withdrawn Submission_

### Official Review · Reviewer_Tmni · 2024-11-03

**Soundness:** 3
**Presentation:** 3
**Contribution:** 2
**Rating:** 3
**Confidence:** 4

**Summary:**

In this paper, the authors present IF-MoDGS, a novel approach for scene reconstruction and novel view synthesis (NVS) that eliminates the need for precomputed camera poses and point clouds from Structure-from-Motion (SfM). The method divides the scene into static and dynamic regions, using the static background to estimate camera poses and a specialized dynamic module to handle moving objects. To enhance spatio-temporal consistency, the authors introduce a high-dimensional feature loss and an annealing frequency loss, which improve rendering quality across complex dynamic scenes.

**Strengths:**

The separation of processing static and dynamic regions in a scene is indeed a reasonable approach. However, it's important to note that many existing methods in SLAM have adopted similar strategies for handling static and dynamic elements.

**Weaknesses:**

There are some concerns about the paper:

1. I agree that SfM often struggles to extract accurate camera poses and obtain sparse point clouds in certain scenarios. However, the proposed method has only been tested on the NVIDIA and UCSD datasets, which utilize multi-view camera setups with significant camera angles. In such settings, SfM methods generally perform well in estimating camera poses. If the authors wish to emphasize their contribution to camera pose estimation, it would be beneficial to test the method on casually captured monocular videos, such as those from the DAVIS dataset, where camera angle changes are minimal and pose estimation is more challenging for SfM.

2. What is the accuracy of camera pose estimation compared to the ground truth? How effective and reliable is the proposed pose estimation method?

3. The provided renderd videos exhibit noticeable visual artifacts, such as the car with weird moving speed. Additionally, the videos are rendered from a fixed camera position and angle. How does the rendering performance vary with different viewpoints and positions? Could the authors provide more analysis on this?

4. More importantly, the videos focus solely on the authors' method. Could the authors provide additional video comparisons with baseline methods?

5. Given that the proposed method uses masks, do the baseline methods also utilize masks? How can fair comparisons be ensured when additional information is involved? Furthermore, how critical is the accuracy of the masks? If the mask is inaccurate at the boundaries between static and dynamic regions, what issues might arise?

6. Why are the results for D3DGS so poor? This seems unusual. Could the authors provide more details on the implementation of each baseline method listed in Table 2?

7. Please add citations for the referenced methods in Table 1.

**Questions:**

See above.

---

### Official Review · Reviewer_gVw9 · 2024-11-04

**Soundness:** 2
**Presentation:** 2
**Contribution:** 1
**Rating:** 5
**Confidence:** 5

**Summary:**

This paper proposes a method for dynamic 3D reconstruction without pose initialization using 3D Gaussian Splatting (3DGS). By leveraging color, depth prediction, and motion mask images as inputs, the method decomposes the scene into static and dynamic parts, achieving competitive results on benchmark datasets.

**Strengths:**

- The first dynamic 3DGS approach that operates without pose initialization.
- Superior performance compared to baseline methods.
- The paper is clearly written and easy to follow.

**Weaknesses:**

- The method is overly engineered, utilizing multiple priors (depth maps, motion masks), loss functions, and multi-stage processing. This complexity can obscure the core idea and contribution of the work. Specifically, the method includes separate "initialization" and "reconstruction" modules, despite claiming to be an initialization-free approach. Why can’t these modules be unified?
- Limited novelty: The approach consists of existing modules. The static initialization component is essentially the same as COLMAP-free 3DGS, while the warp field parameterization resembles 4DGS. The method simply breaks down the problem into smaller parts, supported by monocular predictions.
- The method heavily depends on Mask R-CNN’s motion mask, making it susceptible to errors in network predictions. An ablation analysis assessing the sensitivity to motion mask accuracy from different mask inputs would be valuable.

**Questions:**

- Why was the method not tested on the HyperNeRF dataset?
- In Table 1, what does “w/o initial” mean? Does it omit an initialization module?
- Could you explain the motivation behind introducing CSM? I am unclear on why two warping fields (CSM and HexPlane) are necessary. A clearer explanation and an ablation analysis in this regard would be helpful.
- How do you define the static region based on the Mask R-CNN output? For example, in the upper row of Fig. 4, the window could potentially be dynamic as well.

---

### Official Review · Reviewer_Kaz4 · 2024-11-04

**Soundness:** 3
**Presentation:** 3
**Contribution:** 2
**Rating:** 3
**Confidence:** 4

**Summary:**

The paper presents IF-MODGS, a novel approach for reconstructing and rendering dynamic scenes using only monocular camera input without requiring pre-computed camera poses or point clouds. The key contributions include a pipeline that separates static and dynamic regions to estimate camera poses from static backgrounds and generate point clouds for dynamic objects; a Canonical Space Mapper (CSM) that defines a canonical space and applies deformation to link it with different viewpoints and timestamps; enhanced quality in complex spatio-temporal scenes through a combination of high-dimensional feature loss and annealing frequency loss.

**Strengths:**

1. A novel approach to handling dynamic scenes without requiring traditional SfM-based initialization, and proposes to use the separation of static and dynamic components for independent optimization.
2. Enhance the quality of complex spatiotemporal scenes through a combination of high-dimensional feature loss and annealing frequency loss.
3. Well-structured presentation with clear pipeline illustrations.

**Weaknesses:**

The reviewer appreciates the authors' efforts in addressing dynamic scene reconstruction without relying on SfM points and camera parameters as known priors. However, there is considerable room for improvement in the experimental design and comparison of results:

1. First, it appears that Table 1 and Table 2 are redundant, as both seem to display quantitative results on the NVIDIA dataset. Out of all 7 samples in this dataset, the proposed method outperforms the baselines in only 3. Additionally, the quantitative results for the UCSD dataset are missing from the paper.

2. Similar to RoDynRF, which employs two separate neural radiance fields to represent static and dynamic regions of a scene, the proposed method uses two distinct sets of 3D Gaussians for modeling these regions. The reviewer wonders about the performance of the baseline if the same motion mask were applied, optimizing static and dynamic scenes independently.

3. How does the computational efficiency of the proposed method compare to the baselines in terms of training time and testing speed? This aspect is highlighted as one of the main advantages of using GS.

4. The proposed method depends on the motion mask predicted by Mask R-CNN. How well would the method perform if the motion mask were imperfect? How robust is the motion mask prediction method?

**Questions:**

Please see the above section for my questions and suggestions.

---

### Official Review · Reviewer_fVa6 · 2024-11-04

**Soundness:** 3
**Presentation:** 2
**Contribution:** 2
**Rating:** 5
**Confidence:** 4

**Summary:**

The authors propose a dynamic Gaussian reconstruction method. The proposed pipeline consists of two steps: the initial module and the reconstruction module, which handle static and dynamic components separately. The static initial module initializes point clouds and estimates the pose based on static elements, while the dynamic module focuses solely on initializing dynamic point clouds. In the reconstruction phase, the static reconstruction leverages pose estimation and the point clouds, whereas the dynamic part employs a deformation network for dynamic reconstruction and pose estimation. Finally, both static and dynamic components are combined for the final rasterization.

**Strengths:**

1. the paper is easy to understand.
2. the results show the effectiveness of proposed pipeline.

**Weaknesses:**

1. In the Methods section, the authors should provide succinct descriptions of the methods employed, such as COGS and monocular depth, rather than solely citing them. Including specific aspects—such as key features/primary steps/crucial settings. This would enhance clarity and allow readers to understand the context and relevance of these techniques within the proposed framework.
2. In line 292, the authors claim that the dynamic point cloud obtained from the process is unstable due to adjustments made with the static scale and shift values. If the results of the dynamic part are indeed affected by an overall incorrect scale and shift, then simply removing outliers (top 5% and bottom 10%) may not adequately address this issue. While outlier removal can eliminate extreme values, it does not rectify the underlying problem of an incorrect overall scale and shift. The authors should discuss any additional steps taken to correct scale and shift for stability or acknowledge this as a limitation if unresolved.
3. In line 299, how are the parameters (r and s) for the Gaussian representation obtained? The authors only describe the point cloud in the previous sections. To improve completeness, the authors should provide the mathematical formulation or algorithm used to derive these parameters from the point cloud data.
4. Figure 3 and the lower part of Figure 1 share the same framework, which is redundant. I recommend removing the lower portion of Figure 1 to streamline the presentation.
5. Considering NeRF is also a good 3D representation, the authors should also compare to the state-of-the-art NeRF-based dynamic methods, such as DynNeRF[1], CTNeRF [2] , DynPoint [3], MonoNeRF [4]… .
6. I have some doubts regarding the novelty of this work. Previous dynamic reconstruction efforts typically involve first segmenting the scene into dynamic and static components, using depth prior knowledge to assist in the process. This paper appears to be more of an engineering implementation, resembling a combination of existing approaches rather than presenting novel research contributions. To clarify, the authors could explicitly articulate the unique technical contributions of their approach and explain how it advances the field beyond engineering implementation.

[1] Chen Gao, Ayush Saraf, Johannes Kopf, and Jia-Bin Huang. Dynamic view synthesis from dynamic monocular video. In Proceedings of the IEEE/CVF International Conference on Computer Vision, pages 5712–5721, 2021.
[2] Xingyu Miao, Yang Bai, Haoran Duan, Yawen Huang, Fan Wan, Yang Long, and Yefeng Zheng.
Ctnerf: Cross-time transformer for dynamic neural radiance field from monocular video. arXiv preprint arXiv:2401.04861, 2024.
[3] Kaichen Zhou, Jia-Xing Zhong, Sangyun Shin, Kai Lu, Yiyuan Yang, Andrew Markham, and Niki Trigoni. Dynpoint: Dynamic neural point for view synthesis. Advances in Neural Information Processing Systems, 36, 2024.
[4] Fengrui Tian, Shaoyi Du, and Yueqi Duan. Mononerf: Learning a generalizable dynamic radiance field from monocular videos. In Proceedings of the IEEE/CVF International Conference on Computer Vision, pages 17903–17913, 2023.

**Questions:**

1. In lines 260–261, the authors mention using Mask R-CNN to extract a motion mask. Given that Mask R-CNN is primarily an object detection and segmentation method, how is it specifically utilized to obtain the static mask in this context?
2. In section 3.2, the authors propose to use the scale and shift estimated from the static part for the dynamic part. If a particular frame contains an excessively large moving part (which are common in object-centroid dataset), resulting in a small static part and insufficient static depth, how does this affect the accuracy of the estimated scale and shift and how to handle this? Given that the same scale and shift are applied to the dynamic part, the motion could complicate optimization. Without accurate scale and shift estimates, how can the authors ensure successful optimization in such scenarios?
3. It is evident that the initial module is crucial, as it provides both point clouds and the initial pose, with the core of this module being optimization based on view consistency. However, the depth information provided for optimization is inconsistent in scale and shift, and the pose estimation begins from scratch, espectically under synamic condition. How easy is it to optimize under these conditions? Is there a risk of falling into local optima during the optimization process?
4. In lines 294–296, the authors mention using 5% and 10% thresholds. Why were these specific percentages chosen? They appear intuitive. Additionally, what criteria are used to sort the depth values within a frame to derive the 5% and 10% results?
5. How is the proposed CSM network trained? Where does the ground truth for the Gaussian splats (GS) come from?
6. How is the canonical space for each object defined? Can this be applied to each object individually? Additionally, is the static part also transformed into the canonical space? The authors mention using CSM to transfer points to the canonical space, but how are these points converted back to global space before rendering?

---

### Note · Authors · 2024-11-13

**Comment:**

We thank the reviewers for reviewing our paper. After careful consideration, we think our paper is inappropriate for ICLR, and we decided to withdraw our paper.

**Withdrawal Confirmation:**

I have read and agree with the venue's withdrawal policy on behalf of myself and my co-authors.